

# Tiling soil textures for terrestrial ecosystem modelling via clustering analysis: a case study with CLASS-CTEM (version 2.1)

Joe R. Melton[1], Reinel Sospedra-Alfonso[2], and Kelly E. McCusker[2,3]

[1]Climate Research Division, Environment and Climate Change Canada, Victoria, B.C., Canada
[2]Canadian Centre for Climate Modelling and Analysis, Climate Research Division, Environment and Climate Change Canada, Victoria, B.C., Canada
[3]School of Earth and Ocean Sciences, University of Victoria, Victoria, B.C., Canada. Now at University of Washington, Department of Atmospheric Sciences

*Correspondence to:* Joe Melton (joe.melton@canada.ca)

**Abstract.** We investigate the application of clustering algorithms to represent sub-grid scale variability in soil texture for use in a global-scale terrestrial ecosystem model. Our model, the coupled Canadian Land Surface Scheme - Canadian Terrestrial Ecosystem Model (CLASS-CTEM), is typically implemented at a coarse spatial resolution (ca. $2.8° \times 2.8°$) due to its use as the land surface component of the Canadian Earth System Model (CanESM). CLASS-CTEM can, however, be run with tiling

of the land surface as a means to represent sub-grid heterogeneity. We first determined that the model was sensitive to tiling of the soil textures via an idealized test case before attempting to cluster soil textures globally. To cluster a high-resolution soil texture dataset onto our coarse model grid, we use two linked algorithms (OPTICS (Ankerst et al., 1999; Daszykowski et al., 2002) and Sander et al. (2003)) to provide tiles of representative soil textures for use as CLASS-CTEM inputs. The clustering process results in, on average, about three tiles per CLASS-CTEM grid cell with most cells having four or less tiles.

Results from CLASS-CTEM simulations conducted with the tiled inputs (Cluster) versus those using a simple grid-mean soil texture (Gridmean) show CLASS-CTEM, at least on a global scale, is relatively insensitive to the tiled soil textures, however differences can be large in arid or peatland regions. The Cluster simulation has generally lower soil moisture and lower overall vegetation productivity than the Gridmean simulation except in arid regions where plant productivity increases. In these dry regions, the influence of the tiling is stronger due to the general state of vegetation moisture stress which allows a single tile,

whose soil texture retains more plant available water, to yield much higher productivity. Although the use of clustering analysis appears promising as a means to represent sub-grid heterogeneity, soil textures appear to be reasonably represented for global scale simulations using a simple grid-mean value.





## 1   Introduction

Representation of sub-grid variability is a challenging problem in large-scale modelling applications such as Earth System Models (ESMs). ESMs are commonly run at relatively coarse spatial resolutions due to the computational costs associated with these complex models. The terrestrial component of an ESM is also commonly tied to the grid cell size or truncation level of the atmosphere, making it difficult to resolve smaller scale processes. Heterogeneity in precipitation, vegetation, soils (Boone and Wetzel, 1999), topography, and snow cover (Nitta et al., 2014) on spatial scales much smaller than common model grid cells can cause surface fluxes to vary non-linearly across a grid cell. To address this issue, several modelling groups have adopted either a tiling approach, in which each grid cell is divided into a mosaic of tiles with a different tile given for each landscape feature (Avissar and Pielke, 1989; Koster and Suarez, 1992; Essery et al., 2003), or a statistical approach whereby the sub-grid heterogeneity is represented by a probability density function (e.g. Famiglietti and Wood (1994); Pielke et al. (1991); Boone and Wetzel (1999)).

The division of a grid cell into tiles has been attempted for characteristics such as hydrological parameters (Wood et al., 1992; Arora et al., 2001), vegetation present (Molod and Salmun, 2002; Li and Arora, 2012; Melton and Arora, 2014; Ke et al., 2013), land cover change (Landry et al., 2016), precipitation (Arora et al., 2001), elevation (Ke et al., 2013), land surface properties (Avissar and Pielke, 1989), and maximum infiltration (Decharme and Douville, 2005). Many of these reports that tiled the land surface used relatively easily observed, and hence classified, characteristics of the landscape, i.e. vegetation presence/absence, elevation band, vegetation type, etc. To our knowledge the tiling of soil texture has never been reported. We hypothesize that the use of tiled soil textures, rather than taking simple grid-mean values, will result in more realistic model simulations due to the non-linear influence of soil texture on soil hydrological and thermal characteristics. Soil moisture is one of the most important determinants for partitioning of surface fluxes of moisture and heat from net radiation (Shao and Henderson-Sellers, 1996) and precipitation into evapotranspiration and total runoff (Dirmeyer et al., 1999), as well as having a strong influence on vegetation productivity, and the terrestrial carbon cycle, which is of primary interest here. Soil texture influences on plant productivity and community structure should be especially strong in regions with low water availability as has been observed in semi-arid and arid regions (Archer et al., 2002; Hook and Burke, 2000; English et al., 2005). To test our hypothesis, we use clustering algorithms on a recently released high-resolution soil textural dataset. Clustering analysis searches for patterns in datasets based upon their natural structure or grouping. Some examples of clustering analysis in the Earth system sciences includes remote determination of inundated areas (Prigent et al., 2001), land use management zones (Li et al., 2007), ecoregion delineation (Kumar et al., 2011), and fire regimes (Archibald et al., 2013). Given high-resolution soil textural information, a clustering analysis can determine regions of similar soil textures (e.g. river valleys, mountainous slopes) that are smaller than



the size of ESM grid cells. The soil textures of these distinct regions can then be used as a tile to allow representation of this sub-grid heterogeneity in the model without requiring a smaller model grid. Newman et al. (2014) used k-means clustering analysis to determine tiles based, primarily, on the vegetation types present and thus were able to provide the k term (number of clusters) a priori. In clustering soil texture it is desirable to allow the number of soil clusters to vary per grid cell and not be

specified a priori. Our study thus presents two new approaches: the use of a clustering algorithm to determine tiles that does not require a priori information on the number of tiles per gridcell and using tiled soil texture to represent sub-grid heteorogeneity.

In the following sections we, i) introduce CLASS-CTEM and the clustering algorithms (Section 2), ii) evaluate the soil textural tiles found by the clustering algorithms and the resulting CLASS-CTEM outputs against simulations that use a simple grid cell mean soil texture and against an observation-based dataset of gross primary productivity (Section 3), and iii) discuss

these results and give conclusions for the utility of this approach (Section 4).

## 2    Methods

### 2.1    CLASS-CTEM

All simulations were run with the Canadian Land Surface Scheme (CLASS v. 3.6; Verseghy (2012)) coupled to the Canadian Terrestrial Ecosystem Model (CTEM v.2; Melton and Arora (2016)). Together CLASS-CTEM forms the land surface compo-

nent of the Canadian Earth System Model (CanESM), but the simulations presented here were performed off-line to permit easier interpretation. Since CLASS-CTEM uses grid-mean quantities for the coupling of the land surface to the atmosphere -even if the simulation used tiles for the land surface (see Fig. 1 in Melton and Arora (2014)) - the offline simulations general conclusions should be applicable for the coupled model as well.

CLASS operates at a half-hour timestep and performs the land surface water and energy balance calculations. In simulating

the energy balance of the land surface and its interactions with the atmosphere, CLASS uses vegetation attributes such as leaf area index (LAI), canopy mass, rooting depth, and vegetation height. The temperature, and liquid and frozen water contents of three soil layers, of 0.1, 0.25, and 3.75 meters thickness, are determined prognostically. The CLASS parameterization for mineral soils follows that of Cosby et al. (1984) and Clapp and Hornberger (1978) (see Appendix). Organic soils, defined as those cells having an organic matter weight percent greater than 30, are modelled as peat following Letts et al. (2000).

The daily mean soil temperature, soil moisture, and net radiation from CLASS are passed to CTEM at the end of each day. CTEM then calculates the vegetation and carbon dynamics. While most of CTEM operates on a daily timestep, the carbon assimilation from photosynthesis and canopy conductance occur on the CLASS timestep. CTEM calculates the carbon uptake and respiratory costs of nine plant functional types (PFTs) which map directly to four PFTs that CLASS uses. The CLASS PFTs (with corresponding CTEM PFTs in parentheses) are needleleaf tree (needleleaf evergreen and needleleaf deciduous),

broadleaf tree (broadleaf evergreen, broadleaf cold deciduous, and broadleaf drought/dry deciduous), crop (photosynthetic pathway C3 and C4), and grass (C3 and C4). CTEM tracks carbon flow through the leaves, stem and roots of the living plants and the litter and soil C for the detrital pools. For global simulations, CLASS-CTEM is typically run at a grid cell resolution of approximately 2.8° by 2.8° corresponding to a grid cell size of ca. 98 000 $\text{km}^2$ at the equator and ca. 49 000 $\text{km}^2$ at 45°





latitude. CLASS-CTEM has been validated against observation-based datasets from site-level to global (e.g. Peng et al. (2014); Melton et al. (2015); Melton and Arora (2016)).

### 2.1.1 CLASS-CTEM simulation details

The simulations were forced with version 7 of the Climate Research Unit-National Centres for Environmental Protection

(CRU-NCEP) meteorological dataset covering 1901 - 2015 (Viovy, 2016). The meteorological inputs are disaggregated from 6 hourly to half-hourly as laid out in Melton and Arora (2016). To spin up the model, the climate years 1901 – 1925 were repeatedly cycled over until the model reached equilibrium (which is defined by the net biome production simulated to be less than 0.1% of net primary productivity). During the spinup, the land cover and population densities (used by the fire disturbance scheme) were set to that of year 1850 with a global atmospheric $CO_2$ concentration of 284.87 ppm. After the spinup, the

transient simulation ran from 1851 to 2015 with atmospheric $CO_2$ concentrations from Meinshausen et al. (2011). The land cover is derived from the Global Land Cover 2000 (GLC2000) data set for year 2000 (Bartholomé and Belward, 2005). The GLC2000 data is then mapped to the corresponding CTEM PFTs and we use the HYDE v. 3.1 data set (Hurtt et al., 2011) to change crop area with time. The distribution of the C3 and C4 photosynthetic types for the crops and grasses is based upon Still et al. (2003). To run from 1851 - 2015, the climate was cycled over twice from 1901 – 1925 for the years 1851 – 1900,

then allowed to run freely from 1901 - 2015. All simulations had land use change impacts as well as fire disturbance.

### 2.2 High-resolution Soil Texture Dataset

The Global Soil Dataset for use in Earth System Models (GSDE) (Shangguan et al., 2014) is available at 5 arc minute resolution from http://globalchange.bnu.edu.cn/research/soilw (Accessed July 23rd, 2015). GSDE has eight soil layers of depths: 4.5, 9.1, 16.6, 28.9, 49.3, 82.9, 138.3, and 229.6 cm. CLASS-CTEM's requirements for soil textural information include weight

percent sand, clay, and organic matter (OM) as well as soil depth (Verseghy, 2012). We retain CLASS-CTEM's typical soil configuration of three soil layers with layer bottom depths of 10 cm, 35 cm, and 410 cm.

In each GSDE 5 arc minute grid cell, the soil textural values for depths of 4.5 and 9.1 cm were averaged for the clustering of model soil layer 1. Model layer 2 spanning 10 – 35 cm is assumed to be representable by the mean of GSDE layers 16.6 and 28.9 cm and the bottom model layer spanning 35 – 410 cm by the mean of GSDE layers 49.3, 82.9, 138.3, and 229.6 cm.

GSDE does not contain information about soil depth thus the model default soil depth for each grid cell was used (Zobler, 1986). CLASS-CTEM assumes that any part of the ground column below the soil depth is bedrock and simulates water flow only in the soil part of the ground column, while the temperature dynamics are simulated over both the soil and bedrock sections.

### 2.3 Clustering Analysis

Clustering analysis is primarily a tool for database mining in the information sciences but it has had applications in the earth sciences, predominantly for spatial pattern analysis of remote sensing databases (e.g. Prigent et al. (2001); Archibald et al.





(2013)). For the purpose of representing the spatial heterogeneity of soil textures, a clustering analysis algorithm ideally would independently identify the number of clusters without requiring per-grid cell information, beyond the high resolution soil textural information. After a literature survey, we chose the Ordering Points to Identify the Clustering Structure (OPTICS) algorithm (Ankerst et al., 1999; Daszykowski et al., 2002). OPTICS is a density-based clustering algorithm where clusters are

determined to be areas of higher density than the rest of the dataset. Data points in more sparse regions are considered to be noise. Another common clustering algorithm, k-means was not used as it requires the number of clusters as an input parameter and while there are techniques to diagnostically estimate the number of clusters, they are often ambiguous and their results can differ greatly depending on technique chosen (Chiang and Mirkin, 2010).

OPTICS does not directly produce a clustering of the data but rather a hierarchical representation of the data that shows its

density-based structure. A second step, using the algorithm of Sander et al. (2003), then produces the clusters. The OPTICS algorithm searches a neighbourhood of a predefined radius ($\epsilon$) for clusters that contain a minimum number of points ($minPts$). We set $\epsilon$ to infinity and $minPts$ to 5% of the number of data points in the gridcell (sensitivity to the $minPts$ parameter is discussed further in Section 3.3.1). The parameters for the algorithm of Sander et al. (2003) were taken directly from their paper.

### 2.3.1  Application of OPTICS and the Sander et al. (2003) clustering algorithm

The boundaries of each CLASS-CTEM grid cell (1958 total land cells) were used to determine which high-resolution GSDE grid cells would fit within each model cell. Around 1100 GSDE cells fit within a CLASS-CTEM grid cell. From these GSDE cells all points that were not land (lakes, rivers, etc.) were masked out. If the CLASS-CTEM grid cell did not contain more than 100 GSDE cells (which is about 340 km$^2$ at the equator), the CanESM soil textural information was used for that grid cell. This

occurred for four CLASS-CTEM grid cells and is a result of the land mask used by CLASS-CTEM, which is the same as in the CanESM where the exact placement of the land cells is determined somewhat by the needs of the ocean model. The remaining 1954 CLASS-CTEM grid cells were then individually clustered using the OPTICS and Sander et al. (2003) algorithms. The clustering algorithms choose which GSDE grid cells are considered part of the clusters determined for each CLASS-CTEM grid cell. GSDE grid cells that, in soil texture space, are far from regions of higher density are considered noise and excluded

from clusters (see Section 2.3 above), thus the percent of GSDE cells clustered varies between CLASS-CTEM grid cells. We checked the weighted mean of the clusters against the simple mean of the GSDE grid cells for each CLASS-CTEM grid cell and if the difference between them was greater than 10% for sand, clay, or OM, we assigned that cell the simple gridmean soil textures. This 10% limit was exceeded for 53 CLASS-CTEM grid cells, or <3% of the total. The vast majority of the CLASS-CTEM grid cells above this 10% limit were cells where the clustering algorithm had found only one cluster (Fig. A1).

The clustering algorithms were applied to the GSDE grid cells for the first model layer (0 − 10 cm depth). For simplicity, the clustering found in the first layer was then applied to the layers below, i.e. we did not cluster the lower layers separately rather we apply the clustering assignment of each GSDE grid cell from layer one to each of the lower layers. As our study is mostly focused on the determining the impact of sub-grid soil texture on the model outputs, this simple approach is likely sufficient. Each cluster was assigned the same soil depth. Other model inputs like meteorological forcing and prescribed vegetation cover





was the same for each cluster, i.e. each tile within a CLASS-CTEM grid cell had the same PFT fractional coverages on each tile.

# 3    Results and Discussion

## 3.1    Model sensitivity to tiling

We performed a simple test to ascertain model sensitivity to soil texture, the number of soil tiles and if this sensitivity has a saturating number of tiles. For this test, we first ran a simulation of a test grid cell with a soil texture of 50% sand and 50% clay. We then ran different simulations with an increasing number of tiles but with the same proportion of sand and clay percentages for the grid cell weighted mean. Further simulations were i) two tiles each covering half the grid cell (one with 100% sand and the other 100% clay), ii) three tiles each covering a third (one with 100% sand, one with 50% sand and 50% clay, and the

third 100% clay), iii) four tiles each covering a fourth (one with 100% sand, one with 75% sand and 25% clay, one with 25% sand and 75% clay, and the fourth 100% clay), iv) five tiles each covering a fifth (one with 100% sand, one with 75% sand and 25% clay, one with 50% sand and 50% clay, one with 25% sand and 75% clay, and the fourth 100% clay), etc. up until 20 tiles. All tiles were assigned the same vegetation, soil depth and an OM content of 0%. Some example carbon cycle outputs are plotted in Fig. 1 and show drops of slightly less than 10% to almost 20% as we increase from one tile to two. The change

in the carbon outputs from the one-tile simulations then decreases and stabilizes, indicating that the model is not sensitive to a greater numbers of tiles than seven or eight. These results demonstrate the model should indeed be sensitive to tiling of the soil texture and that 'too many' tiles is not necessary, however this test is relatively unrealistic in its choice of soil texture for the clusters so the next section looks at two example grid cells.

## 3.2    Site-level simulations

### 3.2.1    Evaluation of soil textural clusters

Looking first at example grid cells from Sudan and Brazil (Figs. 2 and 3). These sites were chosen because they are from relatively arid regions and therefore soil moisture variations should play a role in the vegetation dynamics. Figures 2 and 3 show the high resolution GSDE grid cell textures for the top CLASS-CTEM soil layer. The clustering algorithm found three clusters for both example grid cells. The weight percent of clay, sand and OM for each cluster can be seen in Figs. 2 and 3

and compared to the original GSDE grid cells. The joint distribution using kernel density estimation for the sand and clay soil contents is also shown. The clustering is able to effectively capture the distinct soil textural regions apparent in both grid cells. Another example cell with a more heterogeneous GSDE soil texture is shown in Fig. A2.

### 3.2.2    Influence on model outputs

The CLASS-CTEM simulated net primary productivity (NPP), heterotrophic respiration (HR), and net ecosystem productivity

(NEP) for the Sudan and Brazil example sites are shown in Figs. 4 and 5, respectively. Model outputs such as NPP, HR and



NEP are important components of the terrestrial C cycle but they are also useful indicators of changes in soil hydrology and thermal regimes since their calculation is influenced by the soil environment as a whole. To investigate the influence of the clustering algorithm, the per tile results are shown alongside the model results taken at the grid-level (as a weighted mean) for the clustering simulation ('Cluster') and the model result if a simple mean of the GSDE soil texture for the CLASS-CTEM

grid cell was used ('Gridmean').

The Sudan grid cell shows relatively large differences between the three tiles determined by the clustering algorithm. The NPP of tile C (with 36% sand, 31% clay and 2% OM) is generally very low which draws down the grid-level NPP for the Cluster simulation, however not greatly as this tile only occupies 8% of the grid cell. The other tiles (A: 91% sand, 4% clay and 1% OM covering 62% of the grid cell and B:67% sand, 15% clay and 1% OM covering 30% of the grid cell) can also differ

greatly especially for HR and NPP. The NPP and NEP is generally higher for the Gridmean simulation while the HR is higher for the Cluster simulation. The different sensitivity of CLASS-CTEM's simulated NPP and HR to each tile's soil texture is at least partially due to the model formulation of these processes. In CLASS-CTEM, GPP, a component of NPP, depends upon a soil moisture stress term that uses the volumetric water content to determine the degree of soil saturation (Equations A5 - A7 in Melton and Arora (2016)) whereas the HR calculation depends on soil matric potential (Melton et al. (2015) and Equations

A33 - A36 in Melton and Arora (2016)). Soil matric potential is calculated as,

$$\Psi = \Psi_{sat} \left( \frac{\theta_l}{\theta_p + \theta_i} \right)^{-b} \tag{1}$$

where $b$ is the Clapp and Hornberger b term (Cosby et al., 1984), $\Psi_{sat}$ is the soil moisture suction at saturation, and $\theta_i$, $\theta_l$, $\theta_p$ is the volumetric ice, liquid, and pore content of the soil layer, respectively. The Cluster grid-level NPP is also slightly more variable than the Gridmean simulation and appears to respond strongly to precipitation changes in this arid grid cell.

The NPP, HR and NEP values at the Brazil test site (Fig. 5) are all higher than the Sudan test site due, in part, to the higher precipitation in the region. The Brazil site's Gridmean simulation generally has similar NPP and NEP to the Cluster simulation but higher HR. This appears to reflect the relatively similar behaviour between the three tiles determined for this location. The largest difference between tiles is for HR which is lower for tile C (43% sand, 36% clay and 3% OM) compared to the sandier tiles, A (91% sand, 4% clay and 1% OM) and B (67% sand, 15% clay and 1% OM).

The differences between the Cluster and Gridmean simulation for these two grid cells indicates that 1) the model is sensitive to soil textural differences, especially for more arid sites, and 2) the influence of clustering soil textures is not uniform and will depend on the conditions unique to each grid cell. We next look at the influence of the clustering on global simulations.

### 3.3 Global simulations

#### 3.3.1 Evaluation of soil textural clusters

On a global scale, the clustering algorithm found, on average, slightly more than three clusters per CLASS-CTEM grid cell (3.1 ±1.5 ; Fig. A3) with few cells having more than five clusters. The global distribution of the number of clusters and the





percent of GSDE grid cells that formed the clusters is shown in Fig. 6. The number of clusters found by OPTICS shows a lower number of clusters in parts of the United States, Europe and China, with higher numbers generally found for South America and part of Africa. There appears to be some dependence between the number of clusters and the original source soil map that was incorporated into GSDE (c.f. Fig. 1 in Shangguan et al. (2014) for the distribution of source maps incorporated into

GSDE). The regions of two original source maps, the General Soil Map of the United States (GSM) and the Soil Database of China for Land Surface Modelling, appear to correlate well with areas of, primarily, single tiles, as determined by the clustering algorithms. The soil textural information from these regions is of higher quality (pers. comm. W. Shangguan, 2016) with more observations contributing to a higher spatial heterogeneity in the original maps incorporated into GSDE. This higher spatial heterogeneity could have lead the clustering algorithms to find no distinct clustering by effectively increasing noise and

obscuring the regions of higher density of soil textural points that indicate a cluster. The GSM map also covers Alaska but given the sparse population and remoteness of the region, the soil textural information could be of poorer quality, and hence lower spatial heterogeneity, for that state. Western Europe could also have higher quality soil data but it is only a sub section of the European Soils Database. To understand if the selection of the $minPts$ parameter caused the predominance of single tiles in these regions, we reduced $minPts$ from 5% to 1% of the number of data points in the gridcell. This did reduce the number

of grid cells with only single tiles in China, the US, and Europe but it also greatly increased the number of tiles everywhere else (Fig. A4). The mean number of clusters found increased to $11.2 \pm 5.2$ with some grid cells having up to 20 cells. Since the model is not sensitive to more than about 7 tiles (see Section 3.1), the original $minPts$ value used appears more appropriate for the majority of the land surface.

The percent of GSDE grid cells that were included in clusters is, on average, $57.0 \pm 20.1$, as not all GSDE soil textural

values are necessarily determined to fall within a cluster (as discussed in Section 2.3.1). The clustering does not, however, appreciably shift the simple grid-mean texture of the CLASS-CTEM grid cells (Figs A5 and A6), i.e. the raw gridmean is similar to the weighted mean of the clusters. The spatial distribution of the percent of GSDE cells clustered is shown in Fig. 6. Areas of northern Eurasia, southeastern Australia and the Prairie region of Canada appear to have lower percentages of GSDE grid cells clustered while areas like northern Africa and the high-latitudes of Canada have higher percentages clustered

although the pattern on the whole is relatively heterogeneous.

### 3.3.2 Influence on model outputs

Global totals of CLASS-CTEM outputs for tiled (Cluster) and grid-mean (Gridmean) simulations for 1996–2015, along with observation-based estimates, are presented in Table 1. The general impact of the clustering integrated over the globe is small. The physical processes such as evaporation and runoff show small differences of around 1% or less. Some variables for the

carbon cycle also show similar small changes with the largest changes occurring for NEP with a 4% difference between simulations and net biome productivity (NBP) with a 5% difference.

Regional differences can be much larger than the modest global differences found between the two simulations. The annual mean simulated soil moisture per soil layer shows some regions to differ by more than 20% between the Cluster and Gridmean simulations (Fig. 7) with more grid cells showing statistically significant differences between the simulations with increasing





soil depth. The Cluster simulation has generally drier soils than the Gridmean simulation, with larger differences visible for arid regions, such as northern Australia, the Middle East, and Mongolia (which have low soil moisture so small changes in absolute amounts will appear as a larger percent change than the same absolute change in a more humid region), while the northern latitudes are wetter for some of the Canadian high north and western Siberia as well as areas of Indonesia and other

parts of Southeast Asia. These patterns are not uniform and can also differ by soil model layer as is the case in the Saharan region where the second layer is generally wetter for the Cluster simulation than the Gridmean, but drier in the third layer. The principle regions of an increase in soil moisture for the Cluster simulation over the Gridmean are in large peatland complexes such as the West Siberian Lowlands, the Hudson's Bay Lowlands, the Mackenzie River delta, and parts of Indonesia. These peatland regions are strongly influenced by the tiling due to the soil OM threshold above which the peat soils parameterization

of Letts et al. (2000) is applied (soil OM >=30%; see Section 2.1). The higher porosity, greater hydraulic conductivity variation with depth, and differing thermal properties all cause greater changes in soil moisture when a grid cell or tile is treated as peat soil as opposed to a mineral soil. The differences in soil moisture appear to be relatively stable throughout the year with relatively little seasonal variation (not shown).

Changes in soil moisture will influence vegetation through changes in water supply and water stress. The mean annual

gross primary productivity (GPP) as simulated by CLASS-CTEM is plotted in Fig. 8. An observation-based estimate (Beer et al., 2010) is provided for reference against the Gridmean simulation GPP. The relative percent difference between the Gridmean and Cluster simulations can be large in arid regions while relatively small elsewhere (again with the understanding that small absolute changes appear relatively larger for areas of low GPP than for the same absolute change in a region of higher productivity). The areas of significant difference are similar to the regions that saw the significant changes in total

soil moisture (Fig. 7) including central Australia, Saharan Africa, and other arid regions. In these arid regions the Cluster simulation produces higher GPP values than the Gridmean simulation. However in these regions the soil moisture in the total soil column was less in the Cluster simulation than the Gridmean simulation (with the exception of Saharan Africa where the second soil layer increased in soil moisture). In non-arid regions, the general effect of the clustering was to slightly lower GPP (resulting in a slightly smaller global GPP ; Table 1). Regions like the peatland complexes that showed significant changes

in soil moisture (Fig. 7) are already moist and rarely experience water stress thus these changes in soil moisture have little impact upon productivity. We investigated the temporal dynamics of GPP in the regions that differed significantly between the Cluster and Gridmean simulations using the dataset of Jung et al. (2011), and found a small improvement in root mean square deviation of the cluster simulation over the gridmean, but it was smaller than the uncertainty of the Jung et al. (2011) dataset which is relatively large in these regions due to sparse observations (e.g. Fig S2 in Beer et al. (2010)) and low productivity.

To understand how lower soil moisture could lead to higher GPP, we selected a grid cell in Australia that saw a large increase in GPP with lower soil moisture (Fig. 9). This grid cell has five tiles determined by the clustering algorithms. Of these tiles, the fifth (tile E; 78% sand, 12% clay, and 1% OM) has much higher productivity than the others, while only occupying 10% of the grid cell. The higher GPP of tile E can be understood by looking at its plant available soil water, $\theta_a$ (m$^3$/m$^3$), which we approximate using the soil's field capacity, $\theta_{fc}$, and wilting point, $\theta_w$,



$$\theta_a = \theta_{fc} - \theta_w \qquad\qquad (2)$$

The GPP formulation of CLASS-CTEM is sensitive to $\theta_a$ (Melton and Arora, 2016) and thus enforces stomatal closing to limit water loss during periods of low moisture availability, resulting in lower productivity. From Fig. 9, we can see tile E, on an annual basis, generally has some plant available water while the other tiles, and Cluster weighted mean (Fig. 9 bottom right) the Gridmean simulation, are commonly strongly water limited resulting in higher GPP for tile E than other tiles and the Gridmean simulation.

The large influence of the clustering in arid regions demonstrates the impact of soil texture when water limitations are important. In these arid regions, the amount of water in the soil column is low and thus soil textural changes that allow greater $\theta_a$ are important, while regions with plentiful moisture are much less influenced by soil texture since water stress is less frequent and the soils generally contain sufficient water for photosynthesis. CLASS-CTEM's pedotransfer functions could also be limiting the influence of the tiling of soil textures. The range in $\theta_a$ for the Cosby et al. (1984) pedotransfer functions, as implemented in CLASS-CTEM, for soils ranging from the most disparate USDA texture classes ('sand' to 'silt' to 'clay') only covers a $\theta_a$ range of 0.08. Using another pedotransfer function may cause CLASS-CTEM to have a greater sensitivity to soil textural changes. For example, the range in $\theta_a$ using the Saxton and Rawls (2006) pedotransfer functions is over double the range of CLASS-CTEM's implementation of Cosby et al. (1984) ($\theta_a$ high to low range of 0.2). Additionally, the Cosby et al. (1984) pedotransfer functions (Fig. A7 and equations A1 - A8), while non-linear, are relatively linear in the regions of most soil textures (Fig. A5). Additionally, the GPP moisture-stress response of CLASS-CTEM could be quite different from another model thus the effects could be somewhat dependant upon the model used.

## 4   Conclusions

Soil texture influences soil hydrology and temperature and is commonly assigned simple mean values across large grid cells. The sub-grid heterogeneity of soil texture can be represented by tiling of the land surface. To test the sensitivity of our model, CLASS-CTEM, to soil texture, we ran simulations of an artificial grid cell with increasing numbers of tiles but the same grid mean soil texture. CLASS-CTEM's carbon cycle outputs were sensitive to the tiling with some outputs changing over 15% and a saturating effect around 7 or 8 tiles. We then used two linked clustering algorithms (OPTICS (Ankerst et al., 1999; Daszykowski et al., 2002) and Sander et al. (2003)) to cluster high resolution soil textures over the relatively coarse CLASS-CTEM model grid (ca. 2.8° by 2.8°). After determining the impact of this tiling at two locations we ran global simulations using tiled soil textures against those with a simple grid mean soil texture. The difference between the two simulations on a global scale were relatively small (<5%) but could be large regionally (>20%). The areas that felt the largest impact due to the soil texture tiling were in arid or peatland regions. Peatland regions were more sensitive to the tiling due to the model parameterization of peatland soils, that is subject to a minimum organic matter limit, and that could be exceeded for single tiles while the simple grid mean remained below the limit. Arid regions saw the largest impact upon GPP due to those regions'





general state of moisture stress on the vegetation whereas the peatland regions generally have abundant soil moisture. Tiles that retained higher levels of plant available water in arid regions would greatly increase GPP causing the grid-level GPP to rise above that simulated when using a simple grid mean soil texture.

In water-limited regions, the inverse-texture hypothesis as put forth by Noy-Meir (1973); Sala et al. (1997) predicts that

coarse-textured soils will support higher above-ground plant net primary productivity than fine-textured soils. This hypothesis has been supported by observations across precipitation gradients (Lane et al., 1998) and we also find this in our simulations for semi-arid and arid regions (Figs. 4, 5, and 9). The role of soil texture is even stronger on plant community composition based on both observations (Lane et al., 1998; Dodd et al., 2002; Dodd and Lauenroth, 1997; Fernandez-Illescas et al., 2001) and modelling studies (Bucini and Hanan, 2007). While our model does have a parameterization for competition between plants for

ground coverage (Melton and Arora, 2016), we do not presently have shrub PFTs. As the major interactions in these regions is between grasses and shrubs, our competition parameterization is unlikely to appropriately capture the dynamics of plant cover due to soil texture as has been reported by observational studies.

While the performance of the tiled grid cells in the arid regions is encouraging, the overall impact of tiling on the terrestrial C cycle is relatively small and thus the use of a simple grid mean soil texture is likely sufficient for most applications. For

large scale applications with a special interest in arid regions, selectively tiling those regions could be useful for capturing the impact of soil heterogeneity on plant productivity.

## 5   Code availability

The python code used for the OPTICS and Sander et al. (2003) algorithms as well as the CLASS-CTEM fortran code is available. Please email the first author for access to the git repository.



## Appendix A: CLASS-CTEM soil pedotransfer functions

Figure A7 demonstrates the non-linear relationships between soil texture and the hydrologic soil state variables. The saturated hydraulic conductivity, $K_{sat}$ (m/s; Fig. A7) is found from the weight percentage sand content, $X_{sand}$ as (Cosby et al., 1984; Verseghy, 2012) :

$$K_{sat} = 7.0556 \times 10^{-6} \exp(0.0352 X_{sand} - 2.035) \tag{A1}$$

while the pore volume, $\theta_p$ (m$^3$/m$^3$; Fig. A7), is also calculated using $X_{sand}$ (Cosby et al., 1984; Verseghy, 2012) ,

$$\theta_p = (-0.126 X_{sand} + 48.9)/100.0 \tag{A2}$$

The soil moisture suction at saturation, $\Psi_{sat}$ (m; Fig. A7), uses $X_{sand}$,

$$\Psi_{sat} = 0.01 \exp(-0.0302 X_{sand} + 4.33) \tag{A3}$$

The hydraulic parameter $b$ (unitless; also called the Clapp and Hornberger B term) is calculated, via the weight percentage clay content, $X_{clay}$ (Cosby et al., 1984; Verseghy, 2012) as,

$$b = 0.159 X_{clay} + 2.91 \tag{A4}$$

The hydraulic conductivity of the soil, $K$ (m/s), is then related to the soil's volumetric liquid water content, $\theta_l$ (m$^3$/m$^3$) via the Clapp and Hornberger (1978) relationship:

$$K = K_{sat}(\theta_l/\theta_p)^{(2b+3)} \tag{A5}$$

In CLASS-CTEM, the field capacity of soil moisture, $\theta_{fc}$ (m$^3$/m$^3$; Fig. A7), is found by setting $K$ in equation A5 to 0.1 mm/d ($1.157 \times 10^{-9}$ mm/s), and then solving for the liquid water content,

$$\theta_{fc} = \theta_p(1.157 \times 10^{-9}/K_{sat})^{1/(2b+3)} \tag{A6}$$

The field capacity of the lowest permeable layer, $\theta_{fc,b}$ (m$^3$/m$^3$), accounts for the permeable depth of the whole overlying soil column, $z_b$ (m), and is found via Soulis et al. (2011)

$$\theta_{fc,b} = \theta_p/(b-1)(\Psi_{sat}b/z_b)^{1/b}[(3b+2)^{(b-1)/b} - (2b+2)^{(b-1)/b}] \tag{A7}$$





At the wilting point, the soil moisture suction, $\Psi_{wilt}$ is set to 150 m. The volumetric water content at the wilting point, $\theta_w$ ($\mathrm{m^3/m^3}$) is then calculated as,

$$\theta_w = \theta_p \left(\Psi_{wilt}/\Psi_{sat}\right)^{1/b} \tag{A8}$$

The thermal regime of the soil is also influenced by soil texture. The volumetric heat capacity of soils in CLASS-CTEM, $C_g$ ($\mathrm{J/m^3/K}$) is derived from the volume fraction (V) and volumetric heat capacity of clay and silt, $C_{fine}$, sand, $C_{sand}$, and organic matter (OM), $C_{OM}$, components of the soil matrix as a weighted average:

$$C_g = \Sigma(C_{sand}V_{sand} + C_{fine}V_{fine} + C_{OM}V_{OM})/(1-\theta_p) \tag{A9}$$

In a similar manner, the soil thermal conductivity, $\tau_g$ ($\mathrm{W/m/K}$), is calculated via a weighted average of the components' thermal conductivities:

$$\tau_g = \Sigma(\tau_{sand}V_{sand} + \tau_{fine}V_{fine} + \tau_{OM}V_{OM})/(1-\theta_p) \tag{A10}$$

Organic soils, defined as those cells having an organic matter weight percent greater than 30, are assigned values of $K_{sat}$, $\theta_p$, $\theta_{fc}$, $\Psi_{sat}$, $b$, $K$, $C_g$, and $\tau_g$ based on peat texture following Letts et al. (2000). The model's first soil layer is assumed to be fibric peat, the second as hemic peat and the bottom soil layer as sapric peat.

*Author contributions.* J. R. M. initiated the study, performed the clustering, ran the model simulations, performed the analysis and wrote the first draft of the manuscript. K. E. M. suggested clustering analysis, provided coding help for the python scripts as well as interpretation and statistical assistance. R. S.-A. helped choose clustering algorithms, compared results to observations, and helped determine CLASSCTEM's sensitivity to the number of clusters. All authors contributed to the final manuscript.

The authors declare that they have no conflict of interest.

*Acknowledgements.* R.S.-A. was supported by a Natural Sciences and Engineering Research Council of Canada (NSERC) Visiting Fellowship. K.E.M. was supported by the CanSISE Network, which is funded by the Natural Science and Engineering Research Council of Canada (NSERC) under the Climate Change and Atmospheric Research (CCAR) programme. We thank Michal Daszykowski for sharing his coding of the OPTICS algorithm and Brian H. Clowers for sharing his porting of this code into Python. We also thank Amy X. Zhang for sharing her Python code implementing the algorithm of Sander et al. (2003). The high-resolution soil textural database was kindly shared by Wei Shangguan. We thank Vivek Arora and Christian Seiler for providing comments to an earlier version of this manuscript.



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



**Table 1.** Global values for CLASS-CTEM model outputs based on simulations using grid-mean soil textures (Gridmean) and tiled simulations derived from the clustering analysis (Cluster). Values are an average over the period 1996 – 2015.

| CLASS-CTEM Output | Cluster | Gridmean | Percent absolute difference | Observation-based estimate |
|---|---|---|---|---|
| Evaporation ($10^3$ km$^3$ yr$^{-1}$) | 78.3 | 78.6 | 0.5 | 83.9±9.9 (Trenberth et al., 2011) |
| Runoff ($10^3$ km$^3$ yr$^{-1}$) | 32.8 | 32.4 | 1.1 | 38.3 (Fekete et al., 2002) |
| Gross primary productivity (GPP) (Pg C yr$^{-1}$) | 133.1 | 133.6 | 0.4 | 123 ± 8 (Beer et al., 2010) |
| Vegetation biomass (Pg C) | 555.00 | 558.46 | 0.6 | 300 - 536 (Forest biomass)[b] |
| Soil carbon mass (Pg C) | 1132.1 | 1119.6 | 1.1 | 1922[c] (Shangguan et al., 2014) |
| Area burnt ($10^4$ km$^2$ yr$^{-1}$ | 484 | 505 | 4.2 | 464 (Randerson et al., 2012) |
| Net ecosystem productivity (NEP) (Pg C yr$^{-1}$) | 4.6 | 4.8 | 4.0 | |
| Net biome productivity (NBP) (Pg C yr$^{-1}$) | 1.0 | 1.1 | 5.0 | 1.0 – 2.5[d] (Le Quéré et al., 2016) |

Percent absolute difference is calculated as abs{100 - [(clustered value / grid-mean value) * 100]}. [a]Value from eight reanalyses for 2002 – 2008, except ERA-40 which was for the 1990s. [b]As summarized in Kauppi (2003). [c]Note this version of CLASS-CTEM does not simulate permafrost C pools. [d]Range of all estimates across 1990-2015 time period.





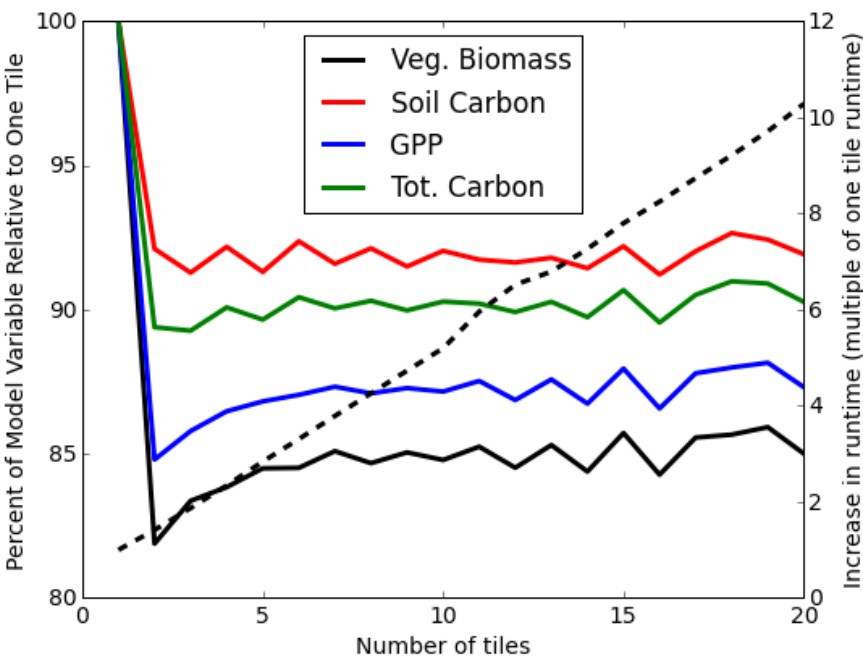

**Figure 1.** Sensitivity test of CLASS-CTEM to the number of tiles (clusters) for a single grid cell. The texture of each tile as the number of tiles increases is described in Section 3.1. GPP is gross primary productivity. All simulations were run until a new equilibrium state was established. The increase in runtime of the model is displayed as a dashed line.





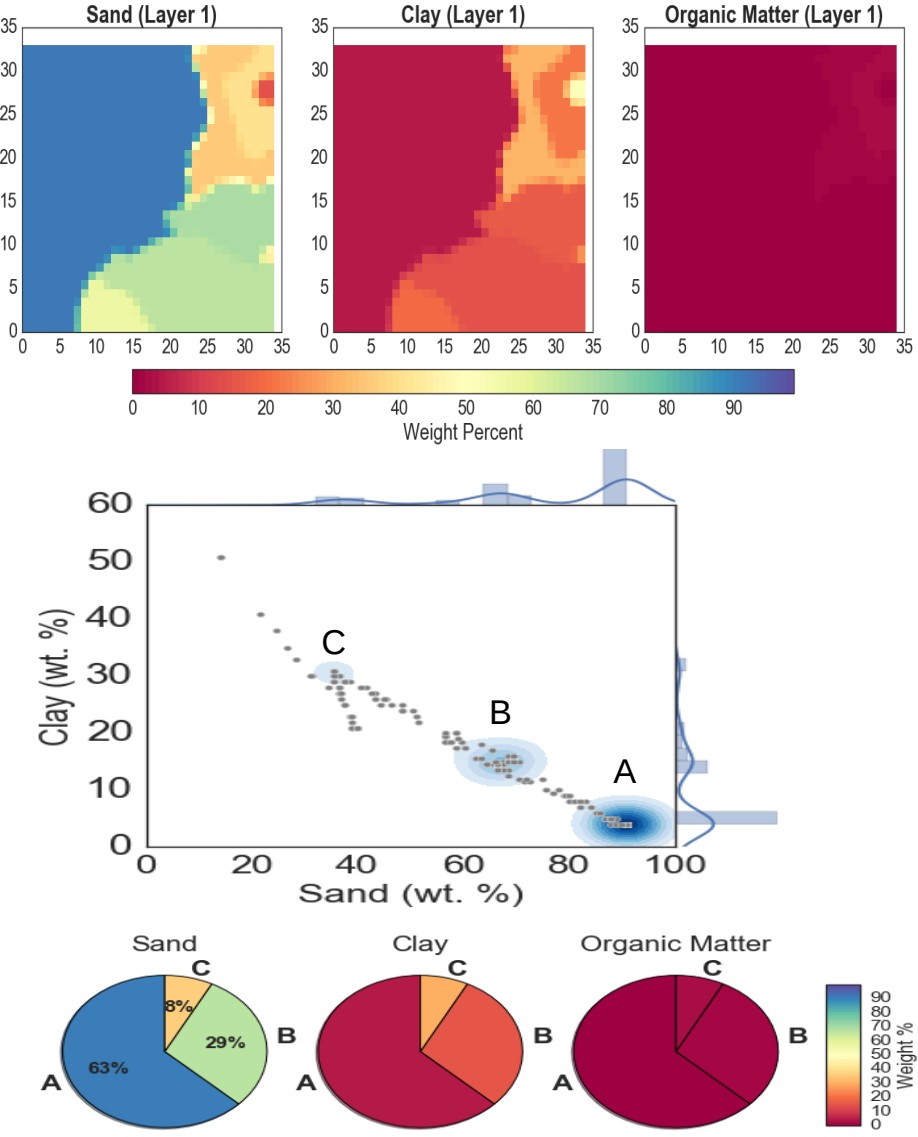

**Figure 2.** Example CLASS-CTEM grid cell located in Sudan (12.6 °N 28.1°E). The top panel shows the GSDE sand, clay, and organic carbon weight percents for GSDE cells within the CLASS-CTEM grid cell. Each GSDE grid cell is 5 arc min by 5 arc min. The numbers on the plot axes are the number of GSDE grid cells along that axis. The joint distribution using kernel density estimation for the soil sand and clay content is shown in the centre panel. The histograms on the axes and the blue colour scaling demonstrate qualitatively the number of GSDE grid cells sharing the similar soil textural space. The clustering algorithm found three clusters (labelled A, B, and C) with a fractional area per cluster and soil texture as shown in the pie charts. The pie charts can be visually referenced to the top panel which uses the same colour scheme, e.g. Cluster A covers 63% of the CLASS-CTEM grid cell with 91% sand, 4% clay and 1% OM. In the scatter plot the label is placed close to the cluster value to help illustrate the cluster relation in sand-clay space.



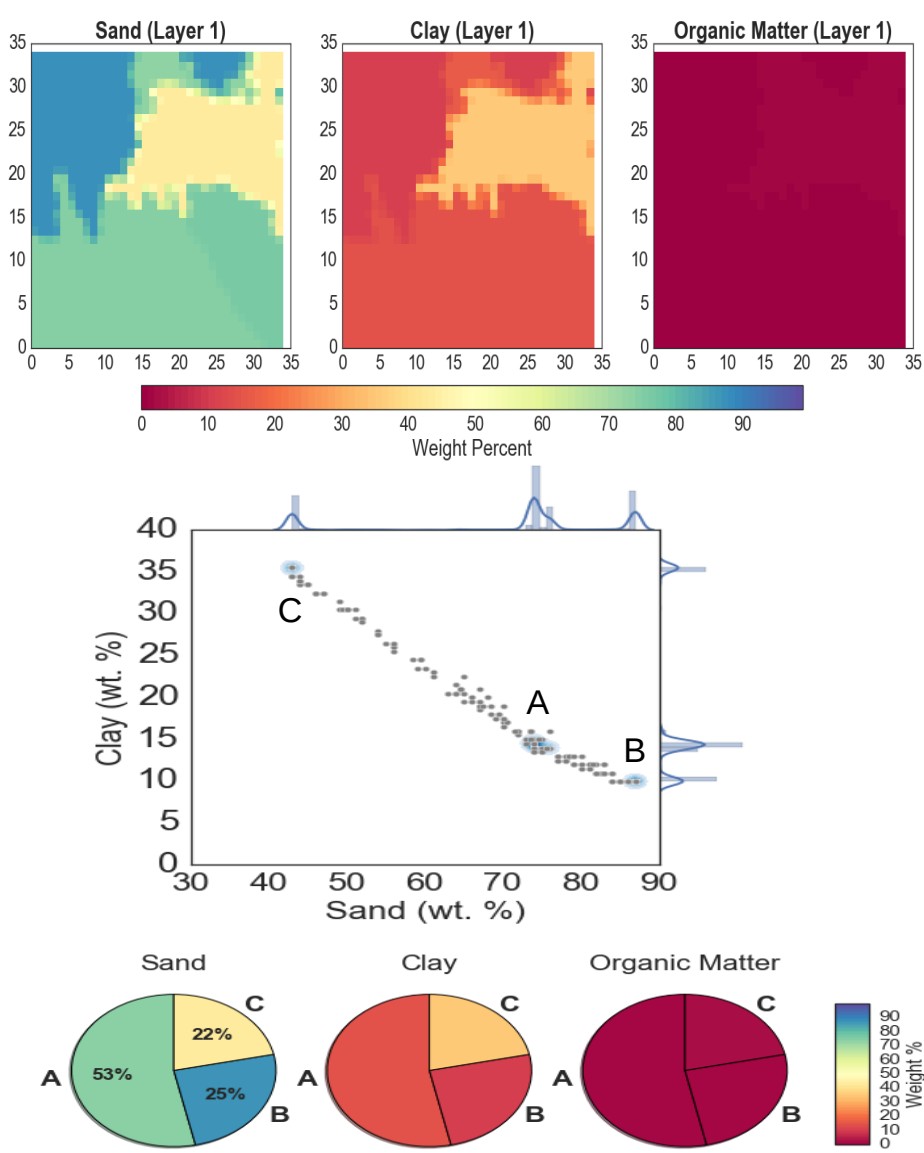

**Figure 3.** Similar to Fig. 2 for a CLASS-CTEM grid cell located in Brazil (9.8 °S 45.0°W).





**Figure 4.** CLASS-CTEM simulated net primary productivity (NPP), heterotrophic respiration and net ecosystem productivity (NEP; NPP – heterotrophic respiration) for the same grid cell in Sudan as Fig. 2. The left column shows the model results per tile with soil textures listed as percent sand/clay/OM along with the tile percent grid cell coverage. The right panel is the model results at the grid-level for the Cluster simulation (weighted mean average of all tiles) and the Gridmean simulation (simple mean of GSDE soil textures). The annual precipitation for this grid cell from CRU-NCEP is included for reference in the upper right plot.







**Figure 5.** Same as Fig. 4 for a grid cell in Brazil (same cell as Fig. 3).





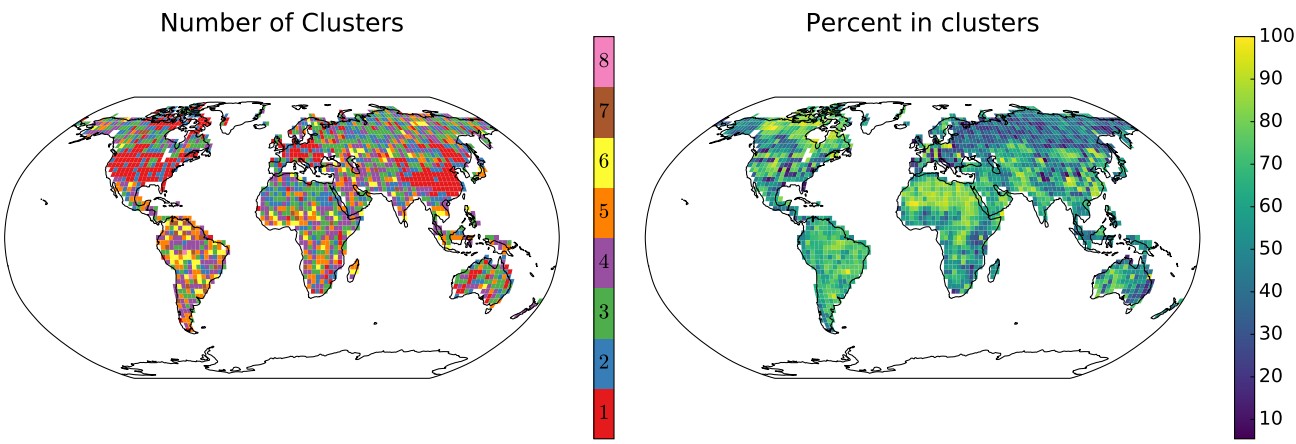

**Figure 6.** Global distributions of the number of clusters (tiles) found per CLASS-CTEM grid cell (left) and the percent of GSDE grid cells clustered per CLASS-CTEM grid cell (right).

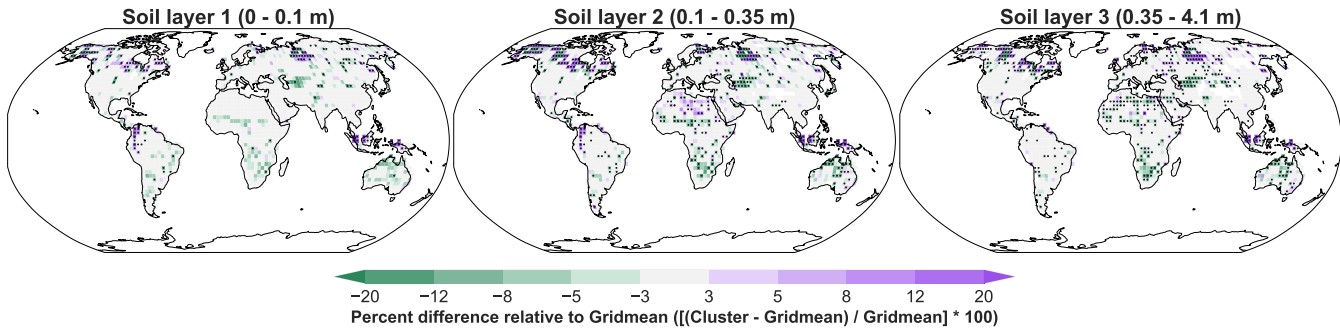

**Figure 7.** Percent difference in soil moisture per CLASS-CTEM soil layer between the Cluster and Gridmean simulations (mean of 1995–2015). Grid cells with soil moisture below $10^{-5}$ kg m$^{-2}$ were masked out to prevent instances of divide by zero and overly large relative differences in regions of very little soil moisture. Positive values indicate the Cluster soil moisture is larger while negative values indicate the Gridmean soil moisture is larger. Dots indicate grid cells that are statistically significant (independent two sample t-test p-level <0.01)





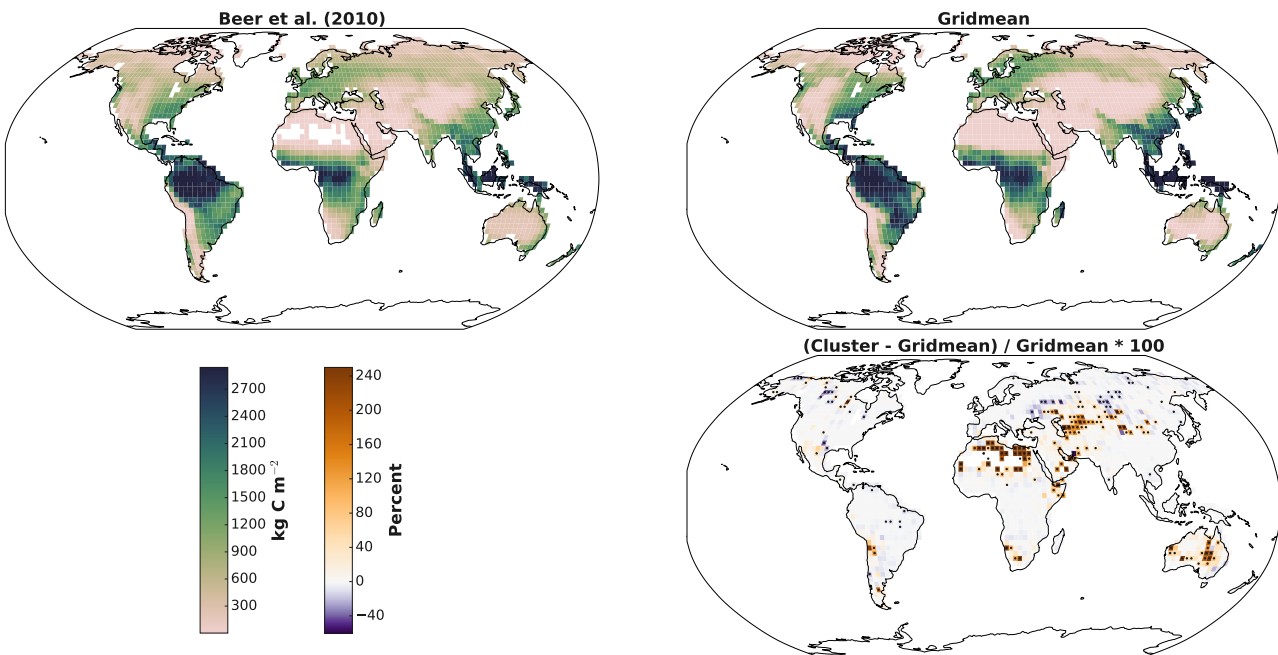

**Figure 8.** Mean annual gross primary productivity (1982 – 2008) from an observation-based dataset (Beer et al., 2010) (top left), the Gridmean simulation (top right), and percent relative difference between the Cluster and Gridmean simulations (bottom right). Note that many of the regions with the largest changes in GPP between the two approaches are also regions with low GPP hence the absolute change in GPP is generally small. Dots indicate grid cells that are statistically significant (independent two sample t-test p-level <0.01). For the areas of significant change in GPP between the Cluster and Gridmean simulations, comparison of Cluster and Gridmean simulations against observations was not significant after accounting for the observational uncertainty.



**Figure 9.** Australian grid cell that has higher GPP for the Cluster simulation than the Gridmean simulation, but lower soil moisture. A measure of the mean annual plant available soil water, scaled so that 1 is field capacity and 0 is wilting point, is calculated for the second model soil layer (0.1 - 0.35 m) and is described in Section 3.3.2. The annual precipitation for this grid cell from CRU-NCEP is included for reference in the upper right plot.





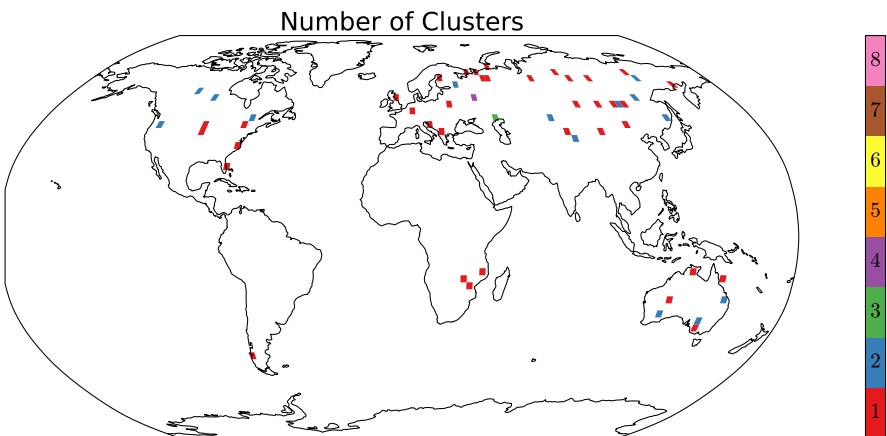

**Figure A1.** Map of the number of clusters for all CLASS-CTEM grid cells where the weighted mean of the clusters was more than 10% different than the simple mean of all GSDE grid cells within a CLASS-CTEM grid cell. These grid cells were then assigned the simple gridmean soil texture values for all simulations (see Section 2.3.1).




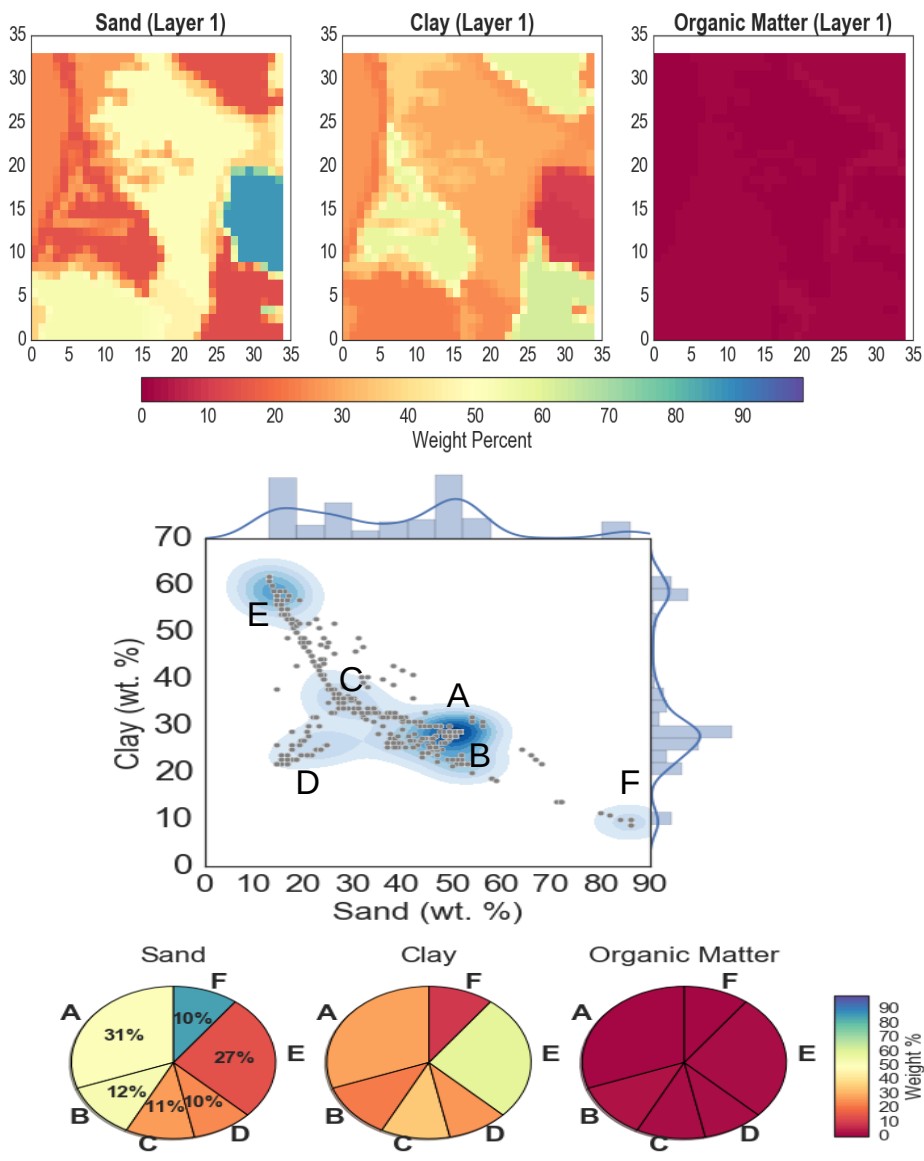

**Figure A2.** Similar to Fig. 2 for a grid cell in Paraguay. This figure demonstrates the clustering performance for a grid cell with a more heterogeneous soil texture. The clustering algorithm found six clusters for this grid cell. The clusters are labelled A through F in the bottom pie chart and middle scatter plot. In the scatter plot the label is placed close to the cluster value to help illustrate the cluster relation in sand-clay space.





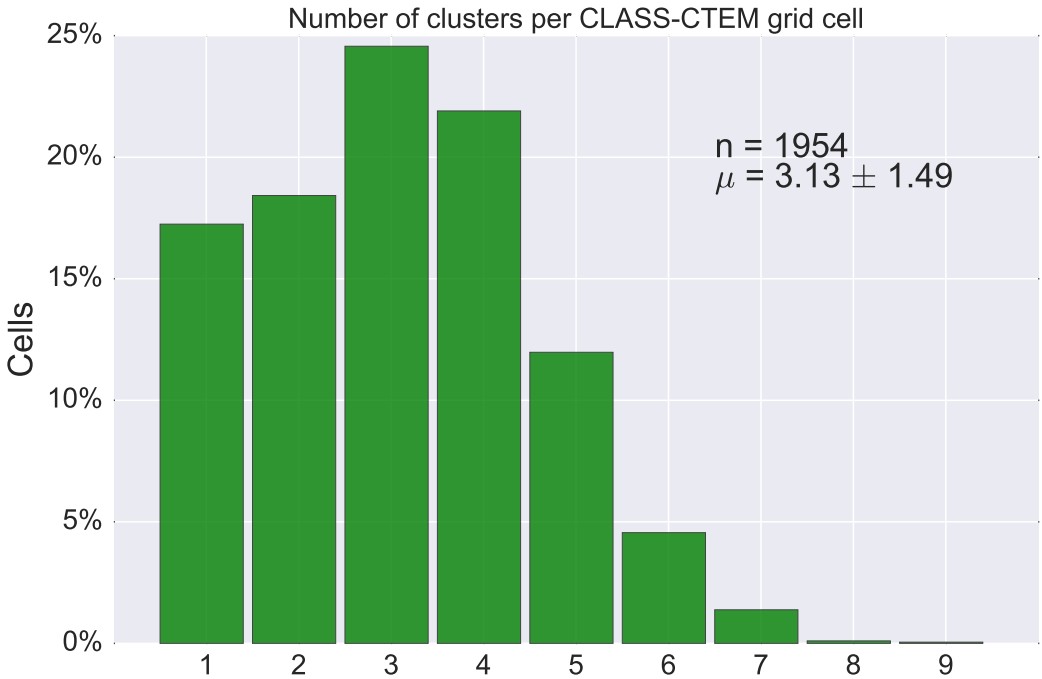

**Figure A3.** Number of clusters determined per CLASS-CTEM grid cell for the GSDE (Shangguan et al., 2014) dataset.

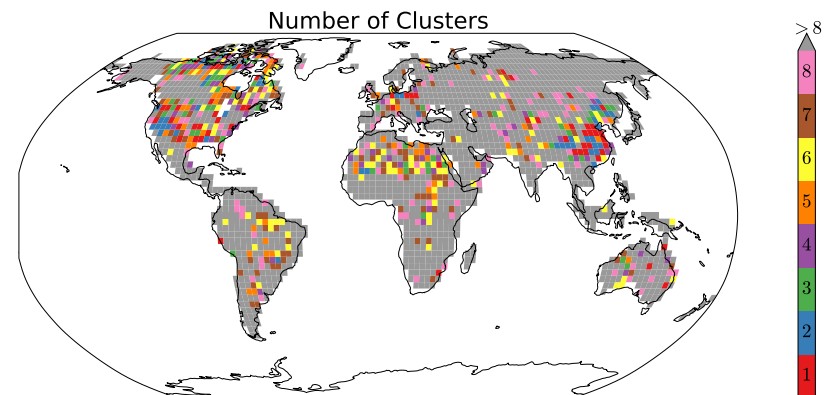

**Figure A4.** Global distributions of the number of clusters (tiles) found per CLASS-CTEM grid cell when $minPts$ (see Section 2.3) is set to 1% of the number of GSDE data points in the CLASS-CTEM gridcell. The gray regions have >8 tiles found by the clustering algorithms.





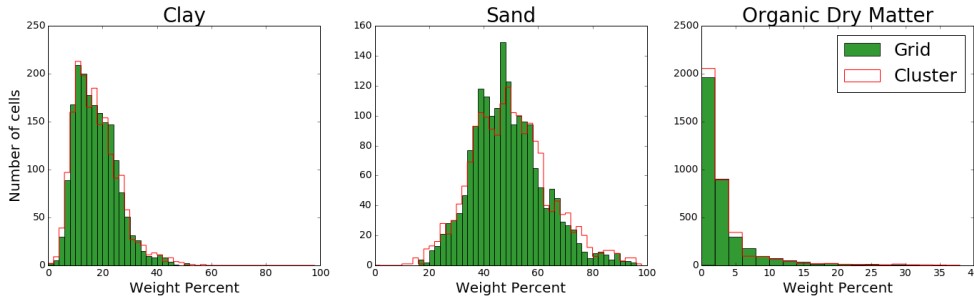

**Figure A5.** Histogram of the mean clay, sand, and organic matter content for CLASS-CTEM grid cells based on the simple mean value of all GSDE cells (green) or the weighted mean of the clusters within a CLASS-CTEM grid cell (red line).

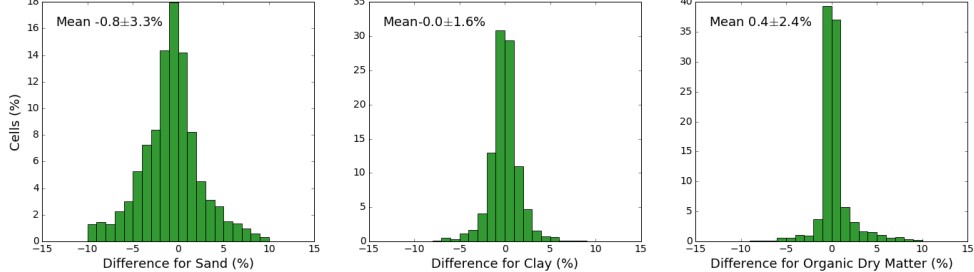

**Figure A6.** Histogram of the difference between the mean clay, sand, and organic matter content for CLASS-CTEM grid cells based on the simple mean value of all GSDE cells and the weighted mean of the clusters within a CLASS-CTEM grid cell.





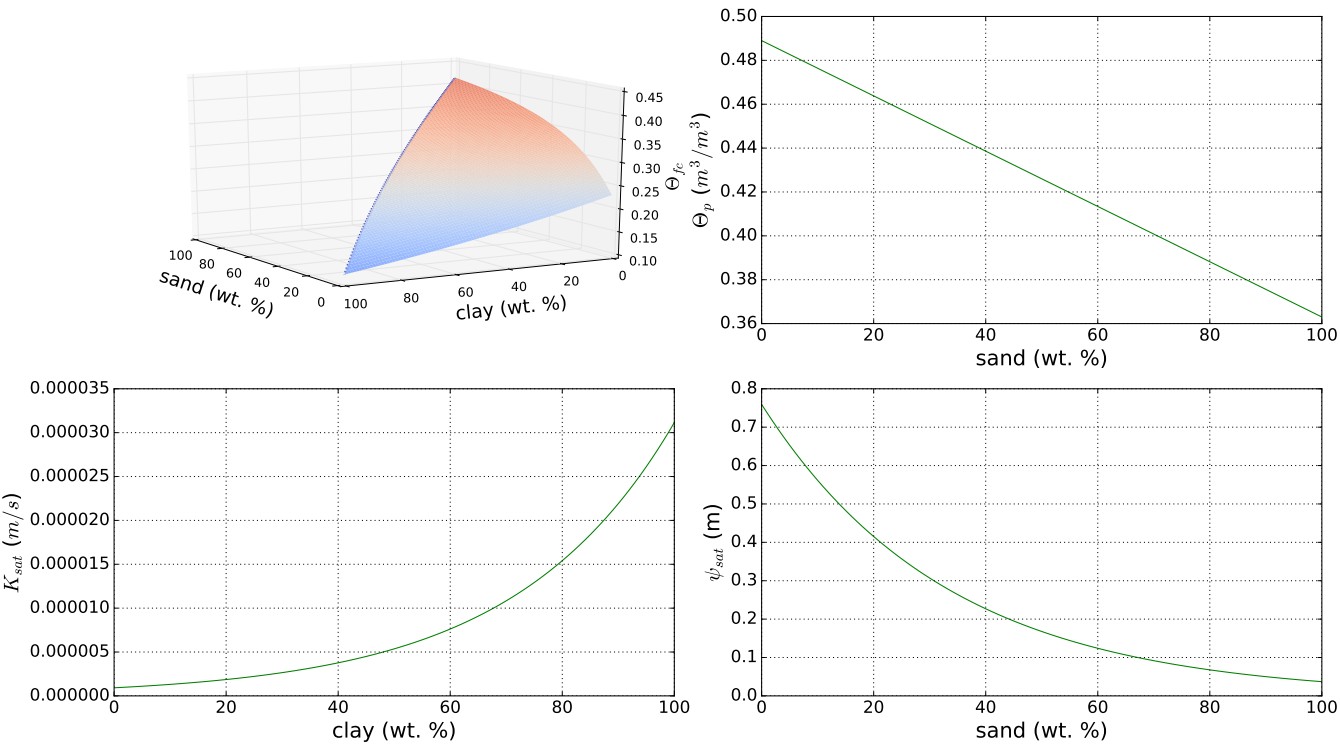

**Figure A7.** Relationships between soil texture and field capacity ($\theta_{fc}$; upper left), pore volume ($\theta_p$; upper right), saturated hydraulic conductivity ($K_{sat}$; lower left), and soil moisture suction at saturation ($\Psi_{sat}$; bottom right) following Cosby et al. (1984) as implemented in Verseghy (2012)