# Peer review of "Tiling soil textures for terrestrial ecosystem modelling via clustering analysis: a case study with CLASS-CTEM (version 2.1)"

_Geoscientific Model Development, 2017_

## Referee Comment (RC1) · H. Zheng (Referee) · 4 Mar 2017

This paper represents valuable efforts of incorporating sub-grid soil heterogeneity in a coupled land surface–ecosystem model, CLASS-CTEM. Instead of a mean value in each large CLASS-CTEM grid cell, several soil texture tiles are identified from a fine-resolution soil texture dataset using a clustering algorithm. The effects of tiling soil texture on modeled soil moisture, net ecosystem exchange, respiration, and carbon storage were examined. Although the authors state that "soil textures appear to be reasonably represented for global scale simulations using a simple grid-mean value," the results could still benefit the modeling community.

The paper is well-written and clearly aligned with the goals of the Geoscientific Model

Development Journal. I recommend its publication with a major revision.

General comments:

1. I am willing to see the modeling results when minPts is set to 1%. The purpose of using soil tiles is to represent spatial heterogeneity. However, by setting minPts to 5%, the most heterogeneous regions (e.g. U.S. and China in Figure 6b) turn out to be the regions with the least numbers of soil tiles (Figure 6a). As a result, I don't think 5% is a reasonable threshold. On the other hand, in your ideal case, there is strong assumption that the fine-scale soil texture distribution is symmetrical. This assumption may not hold in the reality when the soil is heterogeneous.

Some specific comments:

1. P4L4, NCEP = National Centers for Environmental Prediction.

2. P7, L16 and L18. What is "pore content"? Air?

---

## Referee Comment (RC2) · Anonymous Referee #2 · 9 Mar 2017

In this study, the authors investigate a tiling approach for soil textures in CLASS-CTEM with a clustering analysis to identify representing soil textures. It is an interesting study and provides useful information to land modelers although the effect on the terrestrial carbon cycle is relatively small. There are some minor issues that I would like the authors to explain before the manuscript can be accepted for publication.

1) Can you explain more about the treatment of surface heterogeneity in your model? Is the subgrid variability of PFTs represented by a single tile?

2) Why is the sum of the weight percent soil textures less than 100 %?

3) It seems that the effects on C cycle from the sensitivity test (section 3.1) is larger

than the realistic simulations (section 3.2, 3.3). Why?

4) Do the differences depend on the vegetation type?

5) GPP abbreviations need to be explained at line12 on page7.

6) Table 1. Are there any differences in individual components of evaporation (i.e. soil evaporation, transpiration) or runoff (surface runoff and base runoff)?

7) Figure 9. Is  $\Theta$ I in tile E larger than in other tiles? If so, can you explain why? Is it related to the representation of runoff processes?

8) How much does the runtime increase in the global simulation? The variable number of tiles may effectively represent surface heterogeneity while saving computational resources.

---

## Referee Comment (RC3) · Anonymous Referee #3 · 22 Mar 2017

General comments:

The authors examine a clustering algorithm to represent sub-grid heterogeneity of soil texture for a coupled land-surface-ecosystem model (CLASS-CTEM). A newer clustering algorithm is applied to the CLASS-CTEM model soil texture inputs to define the sub-grid tiles in the model. The algorithm has not been previously applied to land-surface/hydrology/ecosystem models for sub-grid tiling and has the potential advantage of determining the number of clusters from the data; no a priori number of clusters needs to be defined. The CLASS-CTEM model shows sensitivity to tiling of soil texture in a synthetic experiment and two grid point simulations. An equilibrium is reached at around 7-8 tiles in the synthetic experiment. They then run global offline simulations.

Overall, clustering by soil texture using the algorithmic parameters discussed in the paper results in roughly 3-4 tiles per grid cell across the globe, with some cells containing up to 7-8+ clusters. The model does not show much sensitivity to ecosystem parameters (e.g. gross or net productivity) across the global uncoupled simulations, except in some arid/semi-arid regions along the margins of deserts.

The paper is very easy to read and logically organized. I think the application of the new clustering algorithm is useful to the community and may have broader applicability; the experiments give insight into this particular clustering method-model combination. However, I have some major concerns. If the editor believes the authors can address them in the allotted time, then it should be acceptable for publication in GMD after revisions, otherwise I would suggest rejection with resubmission.

**Major Comments:**

1) Lines 24-25 on page 2 (Introduction) mentions water balance partitioning (runoff vs ET) and then lines 17-18 on page 3 mention the offline run conclusions should hold for the coupled runs because grid-mean fluxes are passed back to the atmospheric model. However, there is no mention or presentation of any latent or sensible heat fluxes anywhere in the paper. It is very likely that the grid-mean values may be different in the regions of large net ecosystem productivity changes, which would then affect coupled simulations.

I am not expecting fully coupled runs here, but presentation and discussion of flux partitioning changes (or lack thereof) since the primary function of the land-surface model is flux partitioning.

2) Section 3.1: This section needs to be discussed in more detail. What was the vegetation distribution for this synthetic test? Does the vegetation distribution influence the sensitivity? If it does, perhaps another test would be useful. Does this then impact the equilibrium number of clusters?

3) The density-based approach of this clustering algorithm could be problematic. If only 57% of the high-resolution data are being included in clusters on average, you are effectively letting the algorithm determine that 43% of the data may not matter. It is particularly worrisome to see many grid points with clusters that contain only 10-20% of the high-resolution data. Just because the grid-mean texture values and these cluster values agree is not a good enough reason to say the clustering algorithm is successful. The "outliers" or lower-density regions may be important to ecology and/or hydrology. Seyfried et al. (2009) discuss hydrologic hotspots, portions of a catchment that are small but have disproportionate impacts on runoff. The same is seen here in Figure 9 for gross primary production (GPP).

If the clustering is not considering areas of a grid cell that may be small individually (yet add up to a very significant portion of the grid-cell total in many cases), it is possible that some sub-grid variability is being missed (or possibly over-emphasized). For example, what percent of the large grid-cell in Figure 9 is classified on the high-resolution grid? How much of the high-resolution data is similar to (or higher) in sand content than cluster E? If the total area of the large grid cell has more(less) than 10% of the high-resolution points similar to cluster E (from expert evaluation), the grid-cell mean impacts of tiling may be under(over)-estimated.

4) Why is vegetation considered constant across clusters (lines 1-2, page 6)? The GLC2000 dataset could have been mapped to the clusters to have variable PFTs with each tile. Soil texture and vegetation likely co-vary in some or many locations; it would be good to capture that relationship if present.

Minor Comments:

1) Page 5, line 27: Why 10%? Also, this is a 10% change relative to the grid-mean value?

References:

СЗ

Seyfried, M. S., L. E. Grant, D. Marks, A. Winstral, and J. McNamara, 2009: Simulated soil water storage effects on streamflow generation in a mountainous snowmelt environment, Idaho, USA. Hydrol. Processes, 23, 858–873, doi:10.1002/hyp.7211.

---

## Referee Comment (RC4) · E. Blyth (Referee) · 23 Mar 2017

I enjoyed reading this paper - it is extremely well written and logical in its presentation. I think modellers will find themselves informed by it. However, I came away somewhat frustrated not to hear more about the other aspects of the model that might have been affected by this tiling approach. For instance, I would like to see plots of the soil moisture, the evaporation and the drainage (or groundwater recharge) from the models. These land-surface models such as CLASS are increasingly not only used for ecosystem dynamics and soil tiling might make more difference to other applications such as hydrology (droughts, groundwater recharge). This might light up the paper a little especially since the impact on GPP was so small. For instance, you might find a bigger

story to tell with drainage since it is non-linearly related to soil moisture. I think a few extra plots should be enough.

---

## Author Response (AR1)

**Author reply**

**Joe Melton, Reinel Sospedra-Alfonso, Kelly McCusker**

**April 24th 2017**

Dear Editor,

We thank H. Zheng, E. Blyth and the two anonymous reviewers for their time and care in reviewing our paper. We believe that we can address their concerns in our revision. The reviewer comments are repeated below in their entirety with our replies in italic font. We have appended a marked up version of our manuscript showing each change made to make it easy to find the revisions that we have mentioned in the following text.

**Reviewer 1 : H. Zheng**

This paper represents valuable efforts of incorporating sub-grid soil heterogeneity in a coupled land surface–ecosystem model, CLASS-CTEM. Instead of a mean value in each large CLASS-CTEM grid cell, several soil texture tiles are identified from a fineresolution soil texture dataset using a clustering algorithm. The effects of tiling soil texture on modeled soil moisture, net ecosystem exchange, respiration, and carbon storage were examined. Although the authors state that "soil textures appear to be reasonably represented for global scale simulations using a simple grid-mean value," the results could still benefit the modeling community. The paper is well-written and clearly aligned with the goals of the Geoscientific Model Development Journal. I recommend its publication with a major revision.

We thank Dr. Zheng for the positive comments.

General comments:

1. I am willing to see the modeling results when minPts is set to 1%. The purpose of using soil tiles is to represent spatial heterogeneity. However, by setting minPts to 5%, the most heterogeneous regions (e.g. U.S. and China in Figure 6b) turn out to be the regions with the least numbers of soil tiles (Figure 6a). As a result, I don't think 5% is a reasonable threshold. On the other hand, in your ideal case, there is strong assumption that the

fine-scale soil texture distribution is symmetrical. This assumption may not hold in the reality when the soil is heterogeneous.

Reply: A run of the clustering algorithm with a 1% threshold was already included in our original manuscript(MS) (discussed on page 8 line 13 and Figure A4). As can be seen in the figure A4, the number of clusters found in the USA and China does increase as proposed. However the number of clusters also increases broadly across the entire domain.

Figure 1: Number of clusters determined for each CLASS-CTEM gridcell using a 1% threshold. Compare to Fig A3 in the MS.

The mean number of clusters found using a 1% limit increases to over 11 (compared to around 3 using our previous 5% threshold) while the mean percent of GSDE cells clustered (became part of a tile) increased to 65.1  $\pm$  20.7% compared to our clustering using 5% threshold (57.0  $\pm$  20.1%). Generally, clustering using a minPts lower limit of 1% does not result in a gain in information for the model. Clustering with the 1% limit causes a large increase in the number of CLASS-CTEM gridcells with >8 tiles (Fig 1 above and Fig A4 in MS) and, as demonstrated in the MS Fig 1 (and also discussed below in reply to Reviewer 2), the model is generally insensitive to >8 tiles per gridcell. As a result a 5% threshold appears to be reasonable for most locations

For the comment regarding our ideal test case, yes the assumption of the soils being sand and clay mixtures is not likely to be very realistic but the point of this test was twofold: first, to ensure the model was sensitive to soil textural changes, and second, to determine if the model threshold to soil texture changes has some limiting value. Indeed as we can see in the MS's Fig 1 after around 7-8 tiles the model ceases to be sensitive to increased number of tiles. However, in addressing a comment for Reviewer 2 below, we also tested the impact of the grid cells chosen for this test and found that the model sensitivity limit probably differs slightly between gridcells with the driest regions showing sensitivity up to around 12 tiles while wetter tropical and temperate regions are around 7-8. The arid cells with higher sensitivity are arguably less important globally in terms of water and carbon fluxes.

Some specific comments:

1. P4L4, NCEP = National Centers for Environmental Prediction.

Reply: Thanks - corrected.

2. P7, L16 and L18. What is "pore content"? Air?

Reply: Yes, now changed to 'pore (air) content'

**Reviewer 2:** Anonymous**

In this study, the authors investigate a tiling approach for soil textures in CLASS-CTEM with a clustering analysis to identify representing soil textures. It is an interesting study and provides useful information to land modelers although the effect on the terrestrial carbon cycle is relatively small. There are some minor issues that I would like the authors to explain before the manuscript can be accepted for publication.

Reply: Thanks, we are glad you found the work interesting.

1) Can you explain more about the treatment of surface heterogeneity in your model? Is the subgrid variability of PFTs represented by a single tile?

Reply: We took the PFT distribution at the grid cell level and gave each tile the same distribution (see line 34 page 5). For example, if the CLASS-CTEM grid cell had 30% needleleaf evergreen tree, 50% C3 grass and 20% bare ground coverage, each tile would have that same PFT distribution applied to it. We have added an example like this to the text to make sure it is clear. While it is outside the scope of this present study, it would be interesting in the future to try and distribute the PFT coverage to the same tiles as its underlying soil textures, as Reviewer 3 (below) suggests.

2) Why is the sum of the weight percent soil textures less than 100 %?

Reply: The total soil texture is taken to be sand + clay + silt = 100%, thus any remainder is the silt content of the soil. We added this explanation to the revised MS.

3) It seems that the effects on C cycle from the sensitivity test (section 3.1) is larger than the realistic simulations (section 3.2, 3.3). Why?

Reply: The sensitivity test was taken to be an extreme case to help us understand the upper limit for model sensitivity to the tiling. The chosen mineral soil textures - either sand or clay - are representative of the most extreme textural changes the model could experience and thus should produce a large change in the model conditions.

4) Do the differences depend on the vegetation type?

Reply: We have rerun this test for two more sites. The first site is the very dry Sudan site from our MS and the second is a wet tropical site from the Amazon region (-1.40 S 56.25 W). The original site tested is in the N.E. USA (43.3 N 92.8 W). Figure 1 in the appended MS shows the new results. Yes, the magnitude of change does depend on the vegetation type and the general climate of the gridcell. Both the NE USA site and the Amazon site have the highest productivity with a single tile of 50% sand and 50% clay while the Sudan site does much better with two tiles as one tile is then 100% sand (with the other being 100% clay) and that sand tile allows more plant available soil water similar to the Australian test site highlighted in MS Fig 9. Both the N.E. USA and Amazon sites generally have sensitivity up to about 8 tiles while the Sudan site appears sensitive to around 12 tiles. This does seem sensible since our results generally showed a much stronger proportional response to the tiling of soil textures from arid regions. We have added this expanded test along with discussion into the revised MS for Section 3.1. (Please see the appended MS for the changes).

5) GPP abbreviations need to be explained at line12 on page7.

Reply:NPP, NEP, and HR were defined at the bottom of page 6.

6) Table 1. Are there any differences in individual components of evaporation (i.e. soil evaporation, transpiration) or runoff (surface runoff and base runoff)?

Reply: This question was also raised by another reviewer, and is a very valid point. We have now added plots showing the impacts upon the hydrologic cycle to the revised MS (Figs A7 - A9) along with discussion in Section 3.3.2. They are also discussed in more detail in the reply to Reviewer 4 (Eleanor Blyth) below.

7) Figure 9. Is theta\_l in tile E larger than in other tiles? If so, can you explain why? Is it related to the representation of runoff processes?

Reply: No, tile E does not have a larger theta\_l. We have replotted Figure 9 with the soil water content (theta\_l) instead of plant available soil water (Fig 2 below). Tile E has the lowest soil water content while, as can be seen in the original MS Fig 9, it has the highest plant available water.

8) How much does the runtime increase in the global simulation? The variable number of tiles may effectively represent surface heterogeneity while saving computational resources.

Reply: This is an apt point. The offline CLASS-CTEM model timing scales roughly linearly with the number of tiles, but not additively (see dashed line in MS Fig 1; slope around 0.5). As a result, we are interested, as you say, to represent surface heterogeneity in a computationally efficient manner. This is also part of our motivation for why we first tested model sensitivity to the number of tiles. We could easily increase the number of tiles to a very large number but we would not gain any further information for the model and we would greatly increase the computation time. We added some text around this point as it was not emphasized in the original MS version.

Figure 2: Figure 9 replotted with soil water content instead of plant available soil water.

**Reviewer 3 : Anonymous**

General comments:

The authors examine a clustering algorithm to represent sub-grid heterogeneity of soil texture for a coupled land-surface-ecosystem model (CLASS-CTEM). A newer clustering algorithm is applied to the CLASS-CTEM model soil texture inputs to define the sub-grid tiles in the model. The algorithm has not been previously applied to landsurface/hydrology/ecosystem models for sub-grid tiling and has the potential advantage of determining the number of clusters from the data; no a priori number of clusters needs to be defined. The CLASS-CTEM model shows sensitivity to tiling of soil texture in a synthetic experiment and two grid point simulations. An equilibrium is reached at around 7-8 tiles in the synthetic experiment. They then run global offline simulations.

Overall, clustering by soil texture using the algorithmic parameters discussed in the paper results in roughly 3-4 tiles per grid cell across the globe, with some cells containing up to 7-8+ clusters. The model does not show much sensitivity to ecosystem parameters (e.g. gross or net productivity) across the global uncoupled simulations, except in some arid/semi-arid regions along the margins of deserts. The paper is very easy to read and logically organized. I think the application of the new clustering algorithm is useful to the community and may have broader applicability; the experiments give insight into this particular clustering method-model combination. However, I have some major concerns. If the editor believes the authors can address them in the allotted time, then it should be acceptable for publication in GMD after revisions, otherwise I would suggest rejection with resubmission.

*Reply:* Thank you, we are glad you found our paper easy to read and of potential use to the community

Major Comments:

1) Lines 24-25 on page 2 (Introduction) mentions water balance partitioning (runoff vs ET) and then lines 17-18 on page 3 mention the offline run conclusions should hold for the coupled runs because grid-mean fluxes are passed back to the atmospheric model. However, there is no mention or presentation of any latent or sensible heat fluxes anywhere in the paper. It is very likely that the grid-mean values may be different in the regions of large net ecosystem productivity changes, which would then affect coupled simulations. I am not expecting fully coupled runs here, but presentation and discussion of flux partitioning changes (or lack thereof) since the primary function of the land-surface model is flux partitioning.

Reply: This is a good suggestion, thank you for it. We initially chose to focus on the carbon fluxes as that is our primary area of interest but we are happy to expand on the flux partitioning behaviour of the clustered vs. gridmean runs. We now include the differences in sensible (new MS Fig A10) and latent (Fig 3 below) heat fluxes by season as well as the water use efficiency (WUE)(defined as GPP / evapotranspiration; new Fig A11). As in the MS figures, the dots indicate grid cells that are statistically significant (independent two sample t-test p-level

---

## Author Response (AR2)

**Dear Editor,**

We have now uploaded our revision which contains new text addressing the latest comment by Reviewer 3. The new text appears in the Introduction rather than Section 3.3.1 as suggested, as we feel it fits well there where we lay out the logic of choosing clustering analysis as a tool to determine tiles of subgrid soil heterogeneity.

Best regards,

Joe Melton, Reinel Sospedra-Alfonso, and Kelly McCusker.

**Tiling soil textures for terrestrial ecosystem modelling via clustering analysis: a case study with CLASS-CTEM (version 2.1)**

Joe R. Melton1, Reinel Sospedra-Alfonso2, and Kelly E. McCusker2,3

1Climate Research Division, Environment and Climate Change Canada, Victoria, B.C., Canada
2Canadian Centre for Climate Modelling and Analysis, Climate Research Division, Environment and Climate Change Canada, Victoria, B.C., Canada
3School of Earth and Ocean Sciences, University of Victoria, Victoria, B.C., Canada. Now at University of Washington, Department of Atmospheric Sciences

Correspondence to: Joe Melton (joe.melton@canada.ca)

Abstract. We investigate the application of clustering algorithms to represent sub-grid scale variability in soil texture for use in a global-scale terrestrial ecosystem model. Our model, the coupled Canadian Land Surface Scheme - Canadian Terrestrial Ecosystem Model (CLASS-CTEM), is typically implemented at a coarse spatial resolution (ca.  $2.8^{\circ} \times 2.8^{\circ}$ ) due to its use as the land surface component of the Canadian Earth System Model (CanESM). CLASS-CTEM can, however, be run with

- 5 tiling of the land surface as a means to represent sub-grid heterogeneity. We first determined that the model was sensitive to tiling of the soil textures via an idealized test case before attempting to cluster soil textures globally. To cluster a highresolution soil texture dataset onto our coarse model grid, we use two linked algorithms (OPTICS (??) and ?) to provide tiles of representative soil textures for use as CLASS-CTEM inputs. The clustering process results in, on average, about three tiles per CLASS-CTEM grid cell with most cells having four or less tiles. Results from CLASS-CTEM simulations conducted with
- 10 the tiled inputs (Cluster) versus those using a simple grid-mean soil texture (Gridmean) show CLASS-CTEM, at least on a global scale, is relatively insensitive to the tiled soil textures, however differences can be large in arid or peatland regions. The Cluster simulation has generally lower soil moisture and lower overall vegetation productivity than the Gridmean simulation except in arid regions where plant productivity increases. In these dry regions, the influence of the tiling is stronger due to the general state of vegetation moisture stress which allows a single tile, whose soil texture retains more plant available water, to
- 15 yield much higher productivity. Although the use of clustering analysis appears promising as a means to represent sub-grid heterogeneity, soil textures appear to be reasonably represented for global scale simulations using a simple grid-mean value.

The works published in this journal are distributed under the Creative Commons Attribution 3.0 License. This licence does not affect the Crown copyright work, which is re-usable under the Open Government Licence (OGL). The Creative Commons Attribution 3.0 License and the OGL are interoperable and do not conflict with, reduce or limit each other.

©Crown copyright 2016

[revised manuscript text omitted]

| CLASS-CTEM Output                                             | Cluster | Gridmean | Percent absolute | Observation-based estimate                           |
|---------------------------------------------------------------|---------|----------|------------------|------------------------------------------------------|
|                                                               |         |          | difference       |                                                      |
| Evapotranspiration (ET; $10^3 \text{ km}^3 \text{ yr}^{-1}$ ) | 78.3    | 78.6     | 0.5              | 83.9±9.9 ( ? )                                |
| Transpiration (T; $10^3 \text{ km}^3 \text{ yr}^{-1}$ )       | 21.1    | 21.2     | 0.3              | 62±8 (?), 45±4.5 (?)                                 |
| T/ET (%)                                                      | 27.0    | 27.0     | 0.1              | 61±15 ( ? )                                   |
| Runoff $(10^3 \text{ km}^3 \text{ yr}^{-1})$                  | 32.8    | 32.4     | 1.1              | 38.3 (?)                                             |
| Latent heat fluxes (W m -2 )                       | 44.9    | 45.2     | 0.4              | 39 ± 2 (?), 38.5 (?)                                 |
|                                                               |         |          |                  | 37 - 59 (?)                                          |
| Sensible heat fluxes (W m -2 )                     | 25.5    | 24.6     | 3.7              | 41 ± 4 (?), 27 (?)                                   |
|                                                               |         |          |                  | 18 - 57 (?)                                          |
| Water Use Efficiency (g C kg $^{-1}$ water)                   | 1.47    | 1.10     | 32.8             | 1.70 (?)                                             |
| Gross primary productivity (GPP) (Pg C yr -1 )     | 133.1   | 133.6    | 0.4              | 123 ± 8 (?)                                          |
| Vegetation biomass (Pg C)                                     | 555.00  | 558.46   | 0.6              | 300 - 536 (Forest biomass) <math>b</math> |
| Soil carbon mass (Pg C)                                       | 1132.1  | 1119.6   | 1.1              | 1922 c (?)                                |
| Area burnt $(10^4 \text{ km}^2 \text{ yr}^{-1}$               | 484     | 505      | 4.2              | 464 (?)                                              |
| Net ecosystem productivity (NEP) (Pg C yr -1 )     | 4.6     | 4.8      | 4.0              |                                                      |
| Net biome productivity (NBP) (Pg C $yr^{-1}$ )                | 1.0     | 1.1      | 5.0              | $1.0 - 2.5^d$ (?)                                    |

**Table 1.** Global annual values for CLASS-CTEM model outputs based on simulations using grid-mean soil textures (Gridmean) and tiled simulations derived from the clustering analysis (Cluster). Values are an average over the period 1996 – 2015.

Percent absolute difference is calculated as abs{100 - [(clustered value / grid-mean value) \* 100]}. *a* Value from eight reanalyses for 2002 – 2008, except ERA-40 which was for the 1990s. *b* As summarized in ?. *c* Note this version of CLASS-CTEM does not simulate permafrost C pools. *d* Range of all estimates across 1990-2015 time period.

**Figure 1.** Sensitivity test of CLASS-CTEM to the number of tiles (clusters) for three test grid cells. The texture of each tile as the number of tiles increases is described in Section 3.1. GPP is gross primary productivity. All simulations were run until a new equilibrium state was established. The increase in runtime of the model is displayed as a dashed line.